# Au-Embedded and Carbon-Doped Freestanding TiO_2_ Nanotube Arrays in Dye-Sensitized Solar Cells for Better Energy Conversion Efficiency

**DOI:** 10.3390/mi10120805

**Published:** 2019-11-22

**Authors:** Won-Yeop Rho, Kang-Hun Lee, Seung-Hee Han, Hyo-Yeon Kim, Bong-Hyun Jun

**Affiliations:** 1School of International Engineering and Science, Jeonbuk National University, Jeonju 54896, Korea; rho7272@jbnu.ac.kr (W.-Y.R.); dlrkdgns343@naver.com (K.-H.L.); rinmin0616@naver.com (S.-H.H.); rlagydus0203@gmail.com (H.-Y.K.); 2Department of Bioscience and Biotechnology, Konkuk University, Seoul 05029, Korea

**Keywords:** dye-sensitized solar cell, freestanding TiO_2_ nanotube arrays, Au nanoparticles, plasmon, carbon materials

## Abstract

Dye-sensitized solar cells (DSSCs) are fabricated with freestanding TiO_2_ nanotube arrays (TNTAs) which are incorporated with Au nanoparticles (NPs) and carbon materials via electrodeposition and chemical vapor deposition (CVD) method to create a plasmonic effect and better electron transport that will enhance their energy conversion efficiency (ECE). The ECE of DSSCs based on the freestanding TNTAs is 5.87%. The ECE of DSSCs, based on the freestanding TNTAs with Au NPs or carbon materials, is 6.57% or 6.59%, respectively, and the final results of DSSCs according to the freestanding TNTAs with Au NPs and carbon materials is increased from 5.87% to 7.24%, which is an enhancement of 23.34% owing to plasmonic effect and better electron transport. Au NPs are incorporated into the channel of freestanding TNTAs and are characterized by CS-corrected-field emission transmission electron microscope (Cs-FE-TEM) and elemental mapping. Carbon materials are also well-incorporated in the channel of freestanding TNTAs and are analyzed by Raman spectroscopy.

## 1. Introduction

Since the first works by O’Regan and Grätzel in 1991 [1], dye-sensitized solar cells (DSSCs) have been studied due to their high-energy conversion efficiency and low-cost features. They have likewise been expanded to the other solar cells, such as solid-state solar cells and perovskite solar cells. DSSCs have three working parts: one is a working electrode, another is a counter electrode, and the other is an electrolyte. The working electrode mainly consists of dyes, mesoporous TiO_2_ nanoparticles (NPs), and transparent electrode. The electrons are generated from dyes when the dyes get the light energy from the Sun and these electrons are transferred from dyes to mesoporous TiO_2_ NPs. The electrons on the mesoporous TiO_2_ NPs are transported to mesoporous-structured TiO_2_ NPs, which are called to the electron transport, whereas the electrons are re-transferred from mesoporous TiO_2_ NPs to the transparent electrode. After they are circulated, the electrons are transferred to counter-electrodes, which consist of Pt and transparent electrode. During the redox reaction in the electrolyte, the electrons are reduced to dyes. In general, the energy conversion efficiency (ECE) of DSSCs is determined by electron transport and electron generation. 

There are several advantages to DSSCs due to their large surface areas. However, TiO_2_ NPs have several problems regarding improving electron transport because of grain boundaries, defects, and innumerous trapping sites [2,3,4,5]. To overcome these problems in TiO_2_ NPs, TiO_2_ nanotubes are used in DSSCs. TiO_2_ nanotubes, prepared by electrochemical method, have a highly ordered and vertically oriented tubular structure that can be suitable for the electrons’ transport and whose recombination can be reduced compared with the TiO_2_ NPs [6,7,8,9,10,11,12]. Carbon is also one of the better materials for improving electron transport [13]. Carbon has π-π conjugation, which improves the charge separation and electron transport in DSSCs because of good electrical properties. Using plasmon is one of the best ways to improve the electron generation of dyes. Au, Ag, or Al NPs serve as plasmonic materials to enhance the dyes’ light absorption that is related to electron generation [14,15,16,17,18,19,20,21,22,23,24,25,26,27,28,29]. However, it is not easy to incorporate the plasmonic materials and carbon in highly ordered and vertically oriented TiO_2_ nanotubes which are prepared through the electrochemical method. 

In this study, highly ordered and vertically oriented TiO_2_ nanotubes are prepared by means of employing the electrochemical method. They are incorporated with the Au NPs and carbon materials and are fabricated the DSSCs for better energy conversion efficiency.

## 2. Materials and Methods

### 2.1. Preparation of Freestanding TiO_2_ Nanotube Arrays

The Ti plate (99.7%) is purchased from Alfa and the thickness of the Ti plate is 0.25 mm. The Ti plate is done by means of the 1st anodization under the electrolyte whose composition is 0.8 wt% of NH_4_F and 2 vol% of H_2_O in ethylene glycol at a constant applied voltage of 60 V DC for 2 h. After the 1st anodization, TiO_2_ nanotube arrays (TNTAs) on the Ti plate are annealed at 500 °C for 1 h under the ambient condition for crystallinity. After the first anodization, TNTAs on the Ti plate are administered the second anodization under the electrolyte at a constant applied voltage of 30 V DC for 15 min. After the second anodization, the TNTAs on the Ti plate are dipped in the 10% of H_2_O_2_ solution for several minutes to separate the TNTAs from the Ti plate. The bottom layer of separated TNTAs, called as freestanding TNTAs, is removed by the ion milling method for several minutes [12].

### 2.2. Transfer of TNTAs on the Fluorine-Doped Tin Oxide (FTO) Glass

The fluorine-doped tin oxide (FTO) glass is coated with 5 wt% of titanium di-isopropoxide bis(acetylacetonate) in butanol by spin-coater. After the FTO glass is annealed at 500 °C for 1 h under the ambient condition, the crystalline TiO_2_ compact layer is formed on the FTO glass. The TiO_2_ paste is coated on the TiO_2_ compact layer by the doctor blade method, while the freestanding TNTAs are transferred to the TiO_2_ paste. After the substrate is annealed at 500 °C for 1 h under the ambient condition, the adhesion between the TiO_2_ NPs and TNTAs is enhanced [12].

### 2.3. Synthesis of Carbon Materials into the Channel of Freestanding TNTAs

The TNTAs on the FTO glass are placed in a quartz tube furnace with 200 standard cubic centimeters per minute (sccm) of nitrogen gas, with the temperature increased to 450 °C. The 30 sccm of hydrogen gas and 40 sccm of ethylene gas flow into the quartz furnace [13,30,31].

### 2.4. Preparation of Au NPs into the Channel of Freestanding TNTAs

The TNTAs on the FTO glass are dipped in the 10 mM of HAuCl_4_, dissolved in H_2_O and ethanol (50:50, *v*/*v*) solution, and are constantly applied a voltage of 5 V DC. After the electrodeposition method is carried out for Au NPs to be transported into the channel of TNTAs, the Au NPs are treated with TiCl_4_ to prevent redox reaction among the electrolytes in DSSCs [19]. 

### 2.5. Fabrication of DSSCs Based on the TNTAs with Au NPs and Carbon Materials

The TNTAs whose size is 0.5 cm × 0.5 cm and having Au NPs and carbon materials on the FTO glass are immersed in a dye ((Bu_4_N)_2_Ru(dobpyH)_2_(NCS)_2_, N719, solaronix) solution at 50 °C for 8 h and is subsequently sandwiched with counter-electrodes having a 60 μm-thick hot-melt sheet. The electrolyte—0.7 M of 1-butyl-3-methyl-imidazolium iodide (BMII), 0.03 M of I_2_, 0.1 M of guanidium thiocyanate (GSCN), and 0.5 M of 4-tertbutylpyridine (TBP) in acetonitrile and valeronitrile (85:15, *v*/*v*)—is injected between the electrodes.

### 2.6. Instruments

The morphology, thickness, or size of freestanding TNTAs are confirmed by means of a field emission scanning electron microscope (FE-SEM, JSM-6330F, JEOL Tokyo, Japan) with an accelerating voltage in the range of 10 keV and the surface of samples being coated with Pt via an ion sputter. The Au NTs in the channel of freestanding TNTAs is confirmed by Cs-corrected field emission transmission electron microscope (Cs-FE-TEM; JEMARM-200F, JEOL Tokyo, Japan) instrument equipped with X-ray dispersive spectroscopy (EDX) facility, at an accelerating voltage of 200 kV in the Center for University-wide Research Facilities at Jeonbuk National University. The current density-voltage (J-V) characteristics of the DSSCs is confirmed by the electrometer (KEITHLEY 2400, Keithley Instruments, Inc., Cleveland, OH, USA) under AM 1.5 illumination (100 mW/cm^2^) that is provided by a solar simulator (1 kW xenon).

## 3. Results and Discussion

Figure 1 shows the preparation of DSSCs based on the freestanding TNTAs with Au NPs and carbon materials. The TNTAs are prepared by anodization and are detached from a Ti plate that is called to freestanding TNTAs. After TNTAs are separated from the Ti plate, the freestanding TNTAs are transferred to the FTO glass and fixed with the TiO_2_ paste as shown in Figure 1a [32,33,34]. After being sintered at 500 °C, carbon materials are synthesized by chemical vapor deposition as shown in Figure 1b and then Au NPs are incorporated into the channel of freestanding TNTAs by electrodeposition methods as shown in Figure 1c [35,36,37]. Finally, the working electrode, referred to the FTO glass with freestanding TNTAs embedded with Au NPs and carbon materials, assembles the counter electrode, refers to the FTO glass coated with Pt, and injects the electrolyte between the working electrode and counter electrode as shown in Figure 1d.

Figure 2 shows the FE-SEM images of TNTAs. After anodization, the pore size of TNTAs is approximately 100 nm as shown in Figure 2a. After the detachment of TNTAs from the Ti plate, the bottom layer of freestanding TNTAs is observed as shown in Figure 2b. Figure 2c is the Cs-FE-TEM image of Au NPs in the channel of freestanding TNTAs. From Figure 2c, it can be seen that the black lines are the wall of freestanding TNTAs and the black dots are Au NPs whose size is about 40 nm. Figure 2d–f are the elemental mappings of Au, Ti, and O. The green color and yellow color are the Ti and O from the freestanding TNTAs while the red color is the Au from Au NPs. The pattern of green and yellow color is similar to that of the freestanding TNTA wall in Figure 2c while the pattern and position of red color is the same as that of the Au NPs in Figure 2c. From these results, it can be confirmed that the Au NPs are well-prepared in the channel of freestanding TNTAs via the electrodeposition method. 

The crystalline phases of freestanding TNTAs and Au NPs are confirmed by X-ray diffraction (XRD). Figure 3a shows that the crystalline phases of FTO are (110), (101), (200), (221), (310), and (301) planes. The XRD peaks of freestanding TNTAs at 2θ values of 25.35°, 47.95°, 54.05°, 55.15°, 62.80°, 69.05°, 70.40°, and 75.10° correspond to (101), (200), (105), (211), (204), (116), (220), and (215) planes as shown in Figure 3b. The XRD peaks of Au NPs at 2θ values of 38.30°, 44.25°, and 64.45° correspond to (111), (200), and (220) planes as shown in Figure 3c. From the XRD, Au NPs were well-incorporated into the channel of TNTAs by electrodeposition method.

The Au NPs are incorporated into the channel of freestanding TNTAs by means of the electrodeposition method and are also characterized by UV–VIS spectra as shown in Figure 4 [38,39]. The extinction peaks of Au NPs at 5 nm to 80 nm are excited from 514 nm to 550 nm [39,40], while the absorbance band of dye (N719) is 390 nm to 530 nm due to the metal-to-ligand charge transfer. Based on the Cs-FE-TEM, the size of Au NPs is about 40 nm and the stronger peak appears at 520 nm when the extinction peak is under the 390 nm to 530 nm of the absorbance band of dye (N719). For this reason, the electron generation from dye is improved by surface plasmon resonance. 

Carbon materials are synthesized through CVD and analyzed by means of Raman spectroscopy. Figure 5a is the freestanding TNTAs containing the *B_1g_* (397 cm^–1^), *A_1g_* (397 cm^–1^), and *E_g_* (397 cm^–1^) peaks, which indicate that the phase of freestanding TNTAs is anatase structure. Figure 5b illustrates the freestanding TNTAs and the carbon materials after the carbon materials are synthesized through CVD. In Figure 5b, the *B_1g_* (397 cm^–1^), *A_1g_* (397 cm^–1^), and *E_g_* (397 cm^–1^) peaks are also observed, which means that the anatase structure of freestanding TNTAs is not changed after the synthesis of carbon materials. However, the G band and D band peaks are observed at 1575 cm^–1^ and 1340 cm^–1^, which represent the graphite and disorderly network. The D band presents the sp^2^ and sp^3^ sites in the carbon materials. The sp^2^ site particularly means that the carbon materials have π–π conjugation, which improves the electron transport in DSSC [41,42]. For this reason, carbon materials are embedded into the channel of freestanding TNTAs to enhance the electron transport by π–π conjugation. 

Many research groups [24,41,42,43,44,45] have reported that the DSSCs with Au NPs or carbon materials in TiO_2_ NPs improve energy conversion efficiency. This study verified DSSCs with Au NPs and carbon materials in TNTAs. The photovoltaic properties of DSSCs based on the freestanding TNTAs with/without Au NPs or/and carbon materials are measured under the one Sun condition, with the results of the short-circuit density (*J_sc_*), open-circuit voltage (*V_oc_*), fill factor (*FF*), and energy conversion efficiency (ECE, *η*) shown and summarized in Figure 6 and Table 1 and the external quantum efficiency (EQE) shown in Appendix A. The Au NPs are incorporated into the channel of freestanding TNTAs by means of the electrodeposition method [19]. The ECE of DSSC without Au NPs and carbon materials is 5.87%. The ECE of DSSC with Au NPs is 6.57% because of the increasing *J_sc_*. Compared with the *V_oc_* and *FF* of DSSCs without Au NPs and carbon materials, the *V_oc_* and FF of DSSCs with Au NPs are decreased. As Au NPs are incorporated into the channel of freestanding TNTAs, the electron generation of dye is improved by the ‘plasmonic effect’. At this time, some of the electrons are circulated in DSSC for improving *J_sc_* but others are recombined by Au NPs that cause the decreased *V_oc_* and *FF* due to the decreasing Fermi level and electron density of freestanding TNTAs. The ECE of DSSC with carbon materials is 6.57%, which is also increased compared with the ECE of DSSC without Au NPs and carbon materials. In this case, electron transport is improved by carbon materials because the carbon materials have π–π conjugation that furthers the electron transport on the freestanding TNTAs. However, the electron density and Fermi level of freestanding TNTAs are decreased by carbon materials that also decrease the *V_oc_* and *FF*. The ECE of DSSC with Au NPs and carbon materials is 7.24%, which is immediately increased compared with the DSSC with or without Au NPs or carbon materials. In this case, the electron generation and electron transport are improved by plasmonic effect and π-π conjugation from Au NPs and carbon materials. However, the *V_oc_* and FF are lower than those of DSSC without Au NPs and carbon materials because the recombination is increased and electron density and Fermi level are decreased by Au NPs and carbon materials [46]. From the results, it can be confirmed that the Au NPs and carbon materials improve the ECE of DSSC due to the improvement of the *J_sc_* by plasmonic effect and π–π conjugation. 

The electrons are generated in dye (N719) and transferred to the TNTAs. The electrons are transported to the TNTAs and re-transferred to electrode. Successively, the electrons are circulated and reduced to dye by redox process as shown in Appendix A. However, when Au NPs are incorporated in TNTAs, more electrons are generated from dye. Additionally, carbon materials have π–π conjugation that improves not only the electron transport in TNTAs but also the electron transfer between the TNTAs and electrode. The effect of electron transport or transfer of DSSCs based on the freestanding TNTAs with/without Au NPs or/and carbon materials is measured through electrochemical impedance spectra (EIS), and the results *R_s_*, *R_ct1_*, and *R_ct2_* are shown and summarized in Figure 7 and Table 2. The *R_s_* is the series resistance of DSSC, while the starting point of semicircles on the x-axis is the value of *R_s_*. The values of *R_s_* are 7.33 Ω, 7.21 Ω, 7.16 Ω, and 6.89 Ω, corresponding to the DSSCs without Au NPs and carbon materials (a), with Au NPs (b), with carbon materials (c), and with Au NPs and carbon materials (d) as shown in Figure 7. With the Au NPs and carbon materials, the series resistance of *R_s_* exhibits a little decrement due to the increasing electrons by plasmonic effect and π–π conjugation. The *R_ct1_* is the interfacial resistance of FTO/freestanding TNTAs and PT/electrolyte in DSSC. Due to the Au NPs and carbon materials on the freestanding TNTAs, the electrons are well-generated by Au NPs or well-transported by carbon materials. Thus, the interfacial resistance of FTO/freestanding TNTAs is decreased from 1.24 Ω to 1.17 Ω, from 1.24 Ω to 1.16 Ω, or from 1.24 Ω to 1.04 Ω. The *R_ct2_* is the sum of the transfer resistance at the freestanding TNTAs/electrolyte interface and transport resistance in the freestanding TNTAs. With/without Au NPs and/or carbon materials in DSSC, the *R_ct2_* is also decreased from 15.36 Ω to 14.17 Ω, from 15.36 Ω to 12.17 Ω, and from 15.36 Ω to 7.95 Ω, which means that the transfer resistance is also affected by electron generation and electron transport.

## 4. Conclusions

Au NPs were incorporated into the channel of freestanding TiO_2_ nanotube arrays (TNTAs) by means of the electrodeposition method, with carbon materials also synthesized on the freestanding TNTAs through the chemical vapor deposition (CDV) method. The dye-sensitized solar cells (DSSCs) were fabricated with freestanding TNTAs incorporated with Au NPs or/and carbon materials, while the energy conversion efficiency (ECE) was increased from 5.87% to 6.57%, 6.59%, or 7.24%, respectively. The ECE of DSSC based on the freestanding TNTAs with Au NPs was due to the plasmonic effect, while the ECE of DSSC based on the freestanding TNTAs with carbon materials was due to better electron transport on the freestanding TNTAs. Consequently, the ECE of DSSC based on the freestanding TNTAs with Au NPs and carbon materials was improved by plasmonic effect and better electron transport. This research suggested that freestanding TNTAs with Au NPs and carbon materials have great potential for varied kinds of solar cells, batteries, catalysts, water splitters, or sensors.

## Figures and Tables

**Figure 1 micromachines-10-00805-f001:**
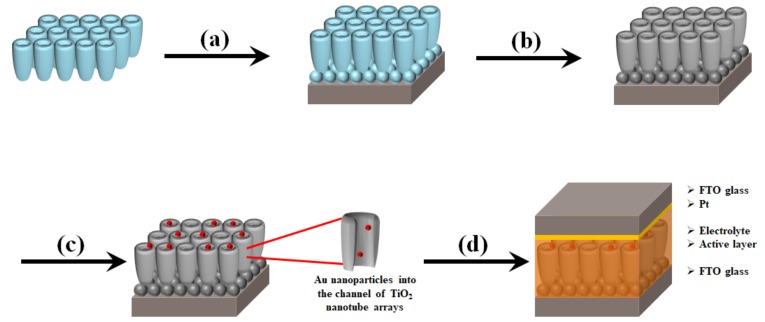
Overall scheme of DSSCs based on the freestanding TNTAs with Au NPs and carbon materials. (**a**) Transfer of freestanding TNTAs on the FTO glass with TiO_2_ paste, (**b**) Synthesis of carbon materials on freestanding TNTAs, (**c**) preparation of Au NPs into the channel of freestanding TNTAs, and (**d**) fabrication of DSSCs.

**Figure 2 micromachines-10-00805-f002:**
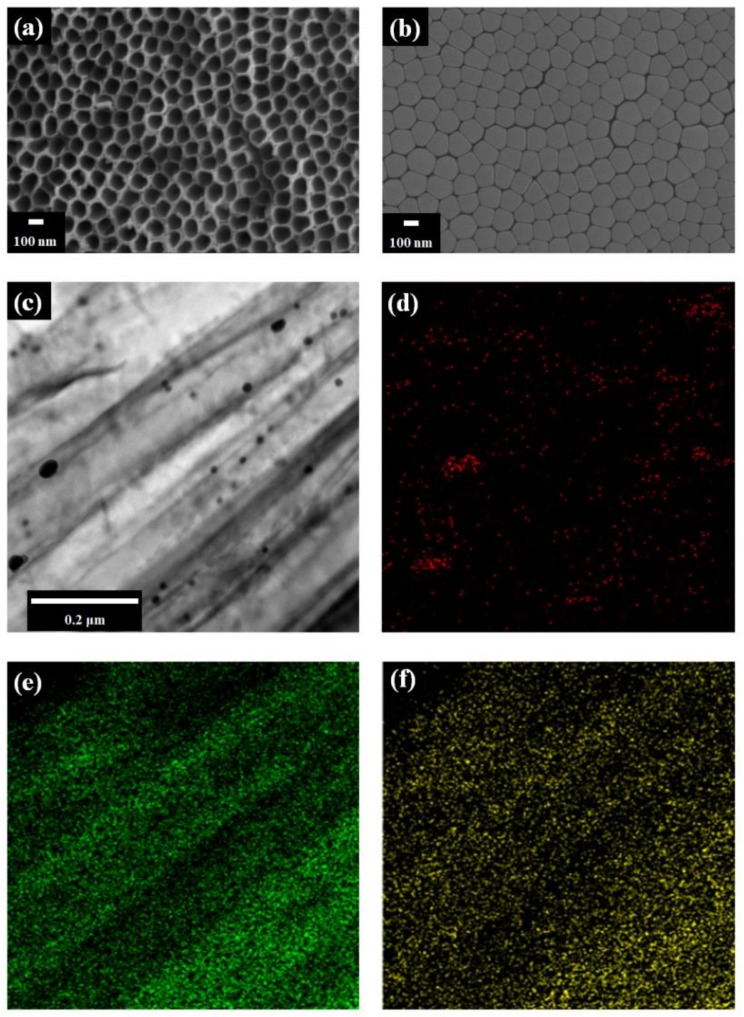
FE-SEM and Cs-FE-TEM images. (**a**) Top of TNTAs and (**b**) bottom layer of freestanding TNTAs by FE-SEM. (**c**) Au NPs by Cs-FE-TEM. Illustrations (**d**), (**e**), and (**f**) are the elemental mappings of Au, Ti, and O.

**Figure 3 micromachines-10-00805-f003:**
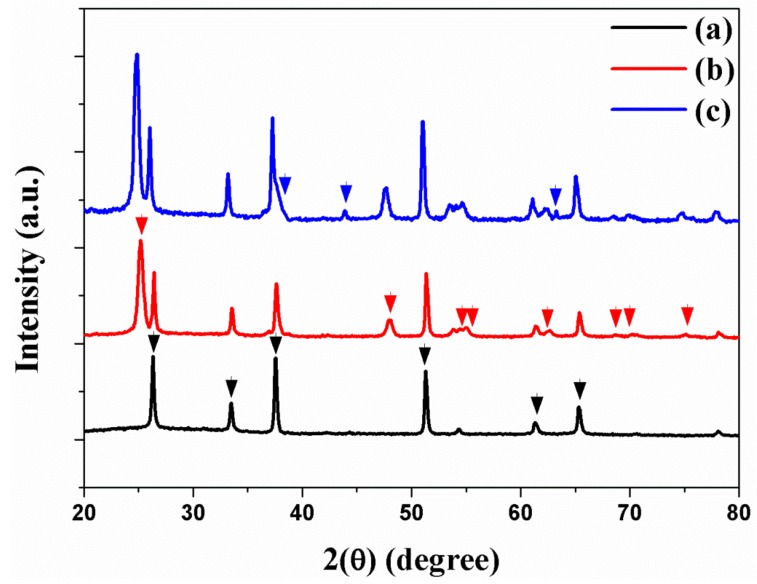
XRD of (**a**) FTO, (**b**) freestanding TNTAs, and (**c**) freestanding TNTAs with Au NPs.

**Figure 4 micromachines-10-00805-f004:**
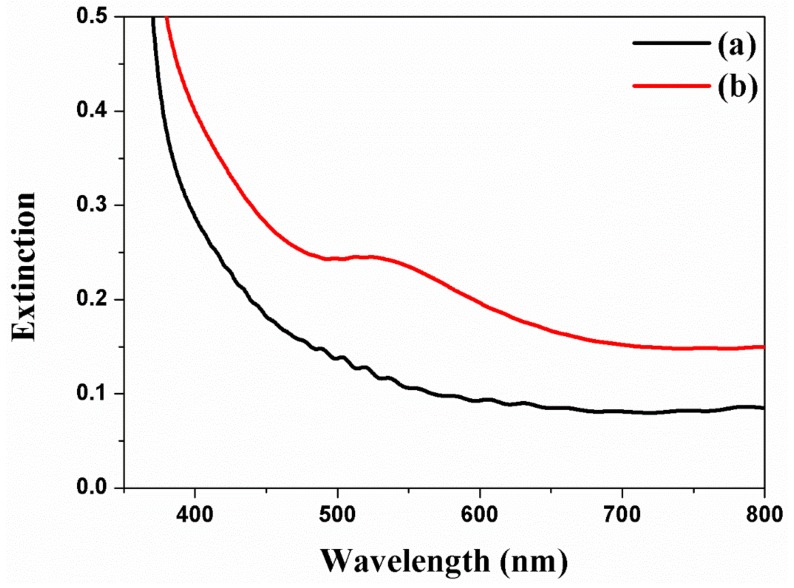
UV-Vis spectra of freestanding TNTAs using Au NPs by means of the electrodeposition method. (**a**) Freestanding TNTAs and (**b**) freestanding TNTAs with Au NPs.

**Figure 5 micromachines-10-00805-f005:**
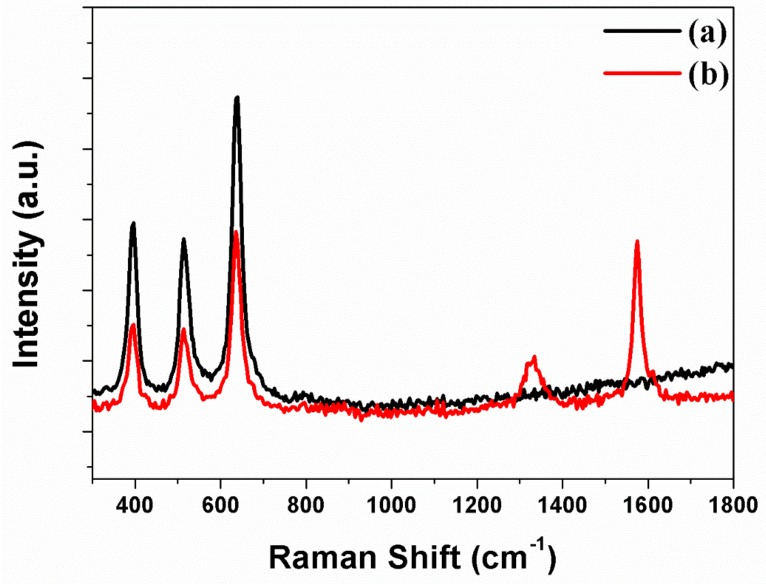
Raman spectra of freestanding TNTAs (**a**) without carbon materials and (**b**) with carbon materials.

**Figure 6 micromachines-10-00805-f006:**
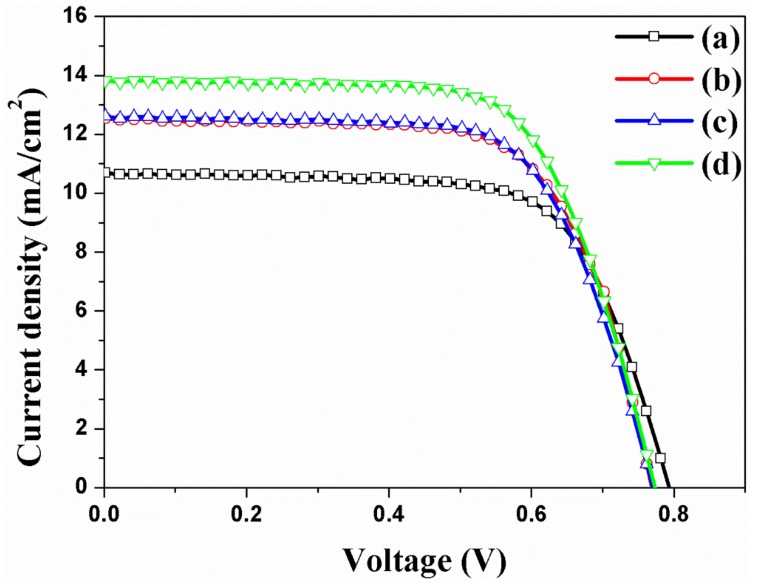
I-V curves of DSSCs based on the freestanding TNTAs (**a**) without Au NPs and carbon materials, (**b**) with Au NPs, (**c**) with carbon materials, and (**d**) with Au NPs and carbon materials.

**Figure 7 micromachines-10-00805-f007:**
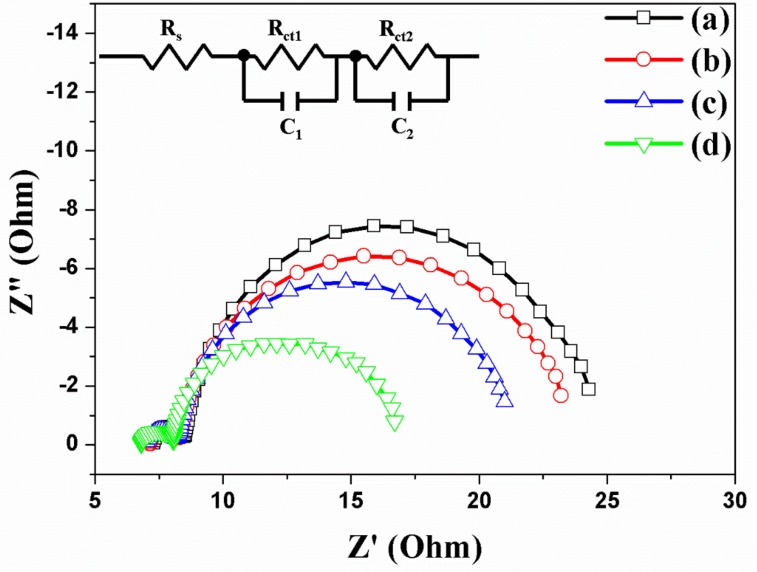
Electrochemical impedance spectra of DSSCs based on freestanding TNTAs (**a**) without Au NPs and carbon materials, (**b**) with Au NPs, (**c**) with carbon materials, and (**d**) with Au NPs and carbon materials.

**Table 1 micromachines-10-00805-t001:** Photovoltaic properties of DSSCs based on freestanding TNTAs with/without Ag NPs and/or carbon materials.

	DSSC Based on the TNTAs	*J_sc_* (mA/cm^2^)	*V_oc_* (mV)	*FF* (%)	*η* (%)
(a)	without Au and carbon	10.64	795	69.4	5.87 ± 0.47
(b)	with Au NPs	12.55	771	68.0	6.57 ± 0.61
(c)	with carbon	12.60	772	67.7	6.59 ± 0.73
(d)	with Au NPs and carbon	13.80	775	67.7	7.24 ± 0.69

**Table 2 micromachines-10-00805-t002:** Fit parameters of electrochemical impedance spectra of DSSCs based on freestanding TNTAs with/without Ag NPs and/or carbon materials.

	DSSC Based on the TNTAs	*R_s_* (Ω)	*R_ct1_* (Ω)	*R_ct2_* (Ω)
(a)	without Au and carbon materials	7.33	1.24	15.36
(b)	with Au NPs	7.21	1.17	14.17
(c)	with carbon materials	7.16	1.16	12.17
(d)	with Au NPs and carbon materials	6.89	1.04	7.95

Note. R_s_: ohmic series resistance; R_1_: sum of small semicircles at high frequency; R_2_: sum of large semicircles at low frequency.

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
