# Peer review of "Au-Embedded and Carbon-Doped Freestanding TiO2 Nanotube Arrays in Dye-Sensitized Solar Cells for Better Energy Conversion Efficiency"

_micromachines, 2019, doi:10.3390/mi10120805_

Round 1

Reviewer 1 Report

Rho et al. introduced Au nanoparticles and carbon materials on TiO2 nanotube arrays (TNTAs) and used them for dye-sensized solar cell photoelectrodes. A significant enhancement (23.34%) in the power conversion efficiency was achieved using plasmonic Au nanoparticles and carbon materials used in the DSSC, as compared to the TNTAs-only based device. This manuscript may draw some interest from the readers in the field of materials science and photovoltaic. However, there are some major issues to be addressed before this manuscript can be accepted for publication.

- The authors must work on re-writing and editing their manuscript. The current version is very hard to read due to the English presentation.

- A very detailed information on the Experimental Section must be provided. Currently, the experimental details are not very clear. What was the active area of the devices? For the Instruments section, the authors should provide the operating conditions of the instruments they used.

- The authors should provide more scientific discussion in the Results and Discussion part.

- In Figure 2, the authors showed elemental mappings of Ti and Au. However, the TNTAs are TiO2. So, the elemental mapping of oxygen (O) must be accompanied with the Ti.

- Not sure what this sentence means "From the XRD of freestanding TNTAs and Au NPs, it can be confirmed that the black line and green line, as shown in Figures 2(c) and 2(d) are the Ti, whereas the black dots and red dots as shown in Figures 2(c) and 2(e) are Au NPs."

- In Figure 3, the Figure caption is "Figure 3. XRD of (a) freestanding TNTAs, (b) freestanding TNTAs with Au, and (c) freestanding TNTAs with carbon materials." However in the text, Figure 3c doesn't seem to be the XRD pattern of TNTA with carbon. Please check and revise it accordingly.

- In Table 1, the authors summarized the photovoltaic parameters of their devices. However it looks like that the authors fabricated only 1 cell per each arhitecture, while it is important to provide consistency results of the fabricated devices. Therefore, the authors are encouraged to make more cells and provide errors bars on their photovoltaic parameters.

- Since authors claiming that the efficiency enhancement is due to the plasmonic effect of Au nanoparticles, the authors should provide external quantum efficiency (EQE) or IPCE measurements on the fabricated devices.

- The authors are encouraged to refer to some important literature in their manuscript:

1. Shapter et al. (Advanced Science 2017, 4 1600504) used carbon nanotubes in photovoltaic devices (perovskite solar cells) and achieved 40% enhancement in the efficiency. The authors used a wide range of experimental analysis and theoretical investigation to explore the internal mechanism of the devices with and without carbon material in the photoelectrode.

2. The same group (Shapter et al., Advanced Science 2015, 2, 1400025) summarized the advances that have been made in the DSSCs with carbon based photoelectrodes.

3. Belcher et al. (ACS Nano 2011, 5, 7108) demonstrated the efficient DSSCs using a plasmonic metal@oxide core-shell structure.

Reviewer 2 Report

In this paper, the authors present a novel method for improving the performance of Dye-sensitized solar cells by plasmonic Au nanoparticle and carbon materials. Significant short circuit current density increase can be observed due to the plasmonic effect and π-π conjugation. I would like to recommend this paper to be published after addressing the following concerns.

Since the plasmonic effect of Au nanoparticles plays important roles in enhancing the current generation, as observed by the authors, it would be better to include the spectra absorption enhancement of the solar cells.

why the authors select Au nanoparticles? Other plasmonic materials, particularly, Al nanoparticles are well demonstrated to lead to a broadband absorption enhancement. I would suggest the authors discuss this in the paper. Some papers regarding Al plasmonic nanostructures as follows could be cited to discuss.

Nanomaterials 6 (6), 95 Journal of Applied Physics 120, 143104 (2016) Nanoscale Horizons 4(3), 601-609 Nano Lett. 2012, 12, 11, 6000-6004 Scientific Reports 3, 2874 (2013)

In the first paragraph of Page 10, the authors state that the Au nanoparticles and/or carbon materials can decrease the recombination in the solar cells and lead to reduced Voc and FF, which could be wrong. Adding these materials inside solar cells introduce more recombination centers and thus increase the recombination. Please clarify this.

In Page 11, the authors state that with the Au NPs and carbon materials, the series resistance of Rs exhibits a little increment. However, the data shows that the series resistance just reduces a little. Please correct this.

Author Response

I have attached the file.

Reviewer 3 Report

In this study, highly ordered and vertically oriented TiO2 nanotubes are prepared by means of employing the electrochemical method. Au NPs and carbon materials are added in the electrode of DSSCs to investigate the change of energy conversion efficiency. This paper could be accepted for publication after minor revised according to the following comments:

1. Au NPs added in the electrode of DSSCs had been widely studied in many research reports. What is novelty of this study? The authors should compare with similar reports by other researchers and discuss in “Results and Discussions” section.
2. In Figure 3, the authors should index the peak of Au NPs and carbon in XRD.
3. What are the transport mechanisms of Au NPs and carbon in the electrode of DSSCs? The authors should describe detail in “Results and Discussion” section.

Author Response

I have attached the file.

Round 2

Reviewer 1 Report

The authors have revised their manuscript carefully and made significant improvement. Therefore, I recommend acceptance for this manuscript.